# Shifting from an expected to an opportunistic pathogen: Comparison of cases of infant late and very late onset group B streptococcal (GBS) infection in a Canadian city over a 27-year period

Isoken Isah[1], Sneha Suresh[1], Gregory J. Tyrrell[2], Manoj Kumar[1], Joan L. Robinson[1]*

1 Department of Pediatrics, University of Alberta, Edmonton, Alberta, Canada, 2 Department of Laboratory Medicine and Pathology, University of Alberta, Edmonton, Alberta, Canada

* jr3@ualberta.ca

## Abstract

### Background

Despite decades of study, the characteristics of infants with invasive group B streptococcus (GBS) late onset disease (LOD) (onset day 7–89 of life) and in particular very late onset disease (VLOD) (after day 89 of life) are not well described.

### Materials and methods

This was a retrospective cohort study of infants hospitalized in four Edmonton hospitals April 1, 1994 through June 30, 2022 with LOD or VLOD GBS invasive disease. Data were collected on demographics, day of onset of infection, clinical manifestations and outcomes.

### Results

There were 115 episodes of LOD in 111 infants of which 50 infants (45%) were preterm. Onset of initial LOD infection was on median day 27 (IQR 19–40.5) of life. All but one infant was bacteremic while 38/111 (34%) had proven and 17/111 (15%) had possible GBS meningitis. Five of 111 (5%) died before hospital discharge with all deaths probably due to GBS. There were 11 episodes of VLOD in 11 infants (8 [73%] preterm) presenting on median day 116 (IQR 103–138) (range 93–207) of life. Three (27%) had GBS meningitis. All 11 survived to discharge. As compared with infants with LOD, those with VLOD were less likely to be born vaginally (n = 3/11 [27%] versus n = 72/111 [65%] p = 0.036), more likely to be mechanically ventilated during their birth hospitalization (n = 5/11 [45%] versus n = 17/111 [15%]; p = 0.039), and more likely to have serious infection with other pathogens during their GBS admission (n = 3/11 [27%] versus n = 4/111 [4%]); p = 0.032). Serotype III accounted for 78% of LOD and 64% of VLOD cases.

which permits unrestricted use, distribution, and reproduction in any medium, provided the original author and source are credited.

**Data availability statement:** Data cannot be shared publicly because of the confidentiality of patient data in a relatively small data set. Data are available from the University of Alberta Institutional Data Access / Ethics Committee for researchers who meet the criteria for access to confidential data. To request the relevant data, researchers may contact 780-492-0302 or Robert Seal at rseal@ualberta.ca.

**Funding:** The author(s) received no specific funding for this work.

**Competing interests:** The authors have declared that no competing interests exist.

## Conclusion

GBS remains a significant cause of infant morbidity and mortality. Infants with VLOD appear to have more complex medical histories prior to GBS infection than those with LOD. There is increasing evidence that GBS may sometimes be an opportunistic pathogen in infants over 89 days of age. This speaks of the need for increased surveillance for VLOD with further research into its epidemiology and underlying immunologic determinants.

## Introduction

*Streptococcal agalactiae,* or group B streptococcus (GBS) is part of rectovaginal flora for approximately 10–35% of women [1]. Carriage in pregnancy can lead to neonatal invasive GBS disease; this was a rare entity until the 1970's when GBS became the leading cause of neonatal sepsis. The reasons for this epidemiologic shift are not clear.

GBS early onset disease (EOD) is transmitted from rectovaginal colonization during labor or at delivery and presents in the first 6 days of life. About 90% of infants present in the first 24 hours of life [2], often with bacteremia and pneumonia. Risk factors for EOD include maternal GBS bacteriuria (an indicator of heavy colonization), chorioamnionitis (as the uterus may contain a high concentration of GBS), prolonged rupture of membranes, preterm delivery (which can be due to symptomatic or subclinical GBS chorioamnionitis) and birth of a previous infant with GBS infection (presumably because the mother does not produce effective GBS antibodies) [1]. Intrapartum antibiotic prophylaxis (IAP) markedly decreases the incidence of EOD [1].

GBS late-onset disease (LOD) occurs day 7–89 and very late onset disease (VLOD) after day 89 of life. Infants with VLOD or LOD typically present with bacteremia with or without a focus [2]. The pathogenesis of GBS acquisition and infection in LOD and VLOD are not established with potential sources being intrapartum transmission [3], the hands of the mother or of other people, breastmilk [1,4], and nosocomial transmission [5] IAP does not change the incidence of LOD [1] so has not been studied for prevention of VLOD.

The objective of this study was to describe and compare the incidence and characteristics of LOD and VLOD in a cohort of neonates in a Canadian city over a prolonged period.

## Materials and methods

This was a retrospective study of a cohort of infants born in Edmonton or St. Albert and hospitalized in one of four Edmonton hospitals April 1, 1994 through June 30, 2022 that had GBS isolated from a sterile site (excluding urine) after day 6 of life. LOD was defined as presentation on day 7–89 while VLOD as presentation on day 90–364 of life. In 1997, the initial Society of Obstetricians and Gynecologists of Canada GBS guideline recommended either rectovaginal screening at 35–37 weeks

gestation with intrapartum antibiotics for all who screened positive or no screening with intrapartum antibiotics for those with risk factors [6]. In 2004 their guideline was revised to recommend only the former approach [7].

Cases were selected via health records search of inpatients discharges up to 12 months of age with discharge ICD9 CM/ICD10CA diagnostic codes that would capture GBS sepsis (S1 Table). Eligibility was confirmed by chart review. Cross referencing of LOD cases was done through a provincial microbiological database of all infants with GBS isolated from a sterile site up to day 89 of life. Cross referencing was not possible for VLOD cases.

Charts were reviewed and data collected on demographics, day of onset of GBS infection, clinical manifestations and outcomes (S2 Table). Serious bacterial infections (defined as bacteriologically proven infections other than skin or mucous membrane infections) with other pathogens that occurred during the GBS admission were recorded data were accessed from 01/07/2016 to 05/08/2025.

Confirmed meningitis was defined as detection of GBS in cerebrospinal fluid (CSF). Possible meningitis was defined as pleocytosis in culture-negative CSF obtained after minimum one dose of antibiotics. CSF pleocytosis was defined as > 20 X $10^6$/L WBCs up to 28 days of age, >, 10 X $10^6$/L WBCs at days 29–56 [8] and > 5 X $10^6$/L WBCs after 56 days of age. For traumatic lumbar punctures, a ratio of 1000 RBCs:1WBC was applied.

For the 2003–2020 cases, microbiologic data but no clinical data were reported in two previous provincial studies [9,10].

Ethics approval was obtained from the University of Alberta Health Research Ethics Board (Pro00059616) who waived the need for parental consent. They approved entry of anonymized data and date of birth into REDCap.

### Data analysis

The data for the characteristics of LOD and VLOD cases was compared to identify any differentiating features among these groups. Chi-square tests were used to compare dichotomous variables, with the use of Fisher's exact test to estimate statistical significance if the expected cell frequency were <5 for any of the cells in 2x2 tables. Data were analyzed using OpenEpi, an open-source software (Available at www.openepi.com). Data on continuous variables are provided as means with standard deviation (SD) or medians with inter-quartile range, depending on the characteristics of the data distribution. Figures were created with the assistance of ChatGPT (OpenAI, San Francisco, CA).

### Capsular polysaccharide (CPS) typing

Neonatal invasive GBS disease is a notifiable disease in Alberta, requiring GBS isolates to be submitted to the Alberta Public Health Laboratory for CPS typing [9]. Prior to January 2017, CPS typing was performed using a double immunodiffusion method with CPS-specific antisera [11]. From 2017 on, a real-time PCR assay was used [12].

## Results

Screening of discharge codes yielded 842 infants of which 122 had confirmed LOD or VLOD.

### Late onset disease (day 7–89 of life)

There were 111 infants with 115 episodes of culture-confirmed GBS LOD (37 with positive blood and CSF cultures, 1 with positive CSF cultures only and 77 with positive blood cultures). Onset of the initial episode of LOD was on median day 27 (IQR 19–40.5) of life; 47 (42%) were male and 50 (45%) were preterm (data on birth gestation missing for 3 cases) (Table 1). At the time of GBS infection, 69 (62%) were receiving breast milk (66 mother's own milk and 3 donor human milk) while 5 (5%) required parenteral nutrition.

At least one CSF was obtained from 108 of the 115 episodes (94%); 38 (35%) were culture-positive (19 obtained prior to antibiotics, 19 after at least one dose antibiotics) and 70 (65%) were culture-negative (22 obtained prior to antibiotics, 42 after at least one dose antibiotics and 6 with unknown timing). For 17 of the CSFs obtained after antibiotics or with

**Table 1. Comparison of LOD and VLOD cases.**

| | LOD (N = 111) | VLOD (N = 11) | P value |
|---|---|---|---|
| Maternal GBS colonization | | | 0.36 |
| Yes | 39 (35%)[1] | 2 (18%) | |
| No | 42 (38%) | 4 (36%) | |
| Unknown | 30 (27%) | 5 (45%) | |
| Previous infant with GBS infection | 2 (2%) | 0 | 0.99 |
| Maternal diabetes | | | 0.99 |
| Gestational diabetes | 9 (8%) | 1 (9%) | |
| Other | 0 | 0 | |
| Corticosteroids prior to birth | 30 (27%) | 3 (27%) | 0.99 |
| Vaginal delivery | 72 (65%)[2] | 3 (27%) | **0.036** |
| Male | 47 (42%) | 3 (27%) | 0.53 |
| Median GA in weeks [IQR] <br> Mean GA in weeks (SD) | 37 [31 –39][3] <br> 34.87 (5.27)[3] | 29 [26.5-36.5] <br> 31.45 (6.35) | 0.08[4] |
| Preterm | 50 (45%)[3] | 8 (73%) | 0.17 |
| Median GA in weeks [IQR] <br> (for preterm infants only) | 30 [27 –34] | 29 [25.5-30.5] | 0.26[4] |
| Mastitis prior to GBS onset | 1 (1%) | 0 | 0.99 |
| Mechanical ventilation during birth hospitalization | 17 (15%) | 5 (45%) | **0.039** |
| NEC | 5 (5%) | 1 (9%) | 0.88 |
| Serious bacterial infection with bacteria other than GBS during GBS admission | 4 (4%)[5] | 3 (27%)[6] | **0.032** |
| Hospital status at GBS onset[7] | | | 0.63 |
| Admitted from home | 77 (69%) | 9 (82%) | |
| GBS during birth hospitalization | 34 (31%) | 2 (9%) | |
| Bacteremic with GBS | 110 (99%) | 9 (82%) | |
| Meningitis | | | |
| Proven GBS meningitis | 38 (34%) | 3 (27%) | |
| Possible GBS meningitis | 17 (15%) | 0 | |
| Day of onset Median [IQR] | 27 (19-40.5)[8] | 116 [101-138] | |

Legend: GBS – group B streptococcus; LOD – late onset disease (day 7–89); NEC – necrotizing enterocolitis; NICU – neonatal intensive care unit; SD – standard deviation; VLOD – very late onset disease (after day 89).

[1]All but 2 got intrapartum antibiotics.

[2]Data missing for 1 case.

[3]Data missing for 3 cases in LOD group.

[4]Comparison using Mann-Whitney U test.

[5]Urinary tract infection (N = 1); coagulase negative staphylococci bacteremia (N = 3).

[6]One case each of urinary tract infection (N = 1); coagulase negative staphylococci bacteremia (N = 1) and meningitis due to *Ureaplasma urealyticus*.

[7]Status at initial GBS LOD episode for the 4 infants who had two episodes.

[8]Day of onset of initial episode of LOD for those with recurrences.

unknown timing, there was CSF pleocytosis so the total number of episodes with confirmed (N = 38) or possible (N = 17) meningitis was 55 (48%). Eight cases had more than one positive CSF culture with these occurring day 1 (N = 1), day 2 (N = 2), day 3 (N = 2) day 4 (N = 2) and day 13 (N = 1) after the initial positive culture.

In addition to bacteremia and/ or meningitis, local sites of infection included adenitis (N = 2), omphalitis (N = 1), cellulitis (N = 3), adenitis and cellulitis (N = 1), and omphalitis and cellulitis (N = 1). Another 7 infants had positive urine cultures for GBS in addition to positive blood or CSF cultures. There were no musculoskeletal infections.

Three of the 111 infants had EOD before the episode of LOD including one infant who went on to have two episodes of LOD. That infant was born at 36 weeks gestation and had GBS bacteremia without a focus on days 6, 19 and 45 of life. No explanation was established but breastfeeding was stopped after the second recurrence as breast milk and infant throat culture grew GBS. Another three infants had two episodes of LOD. A 28-week GA infant had meningitis on day 13 and bacteremia with no focus on day 43 of life (5 days after a 25-day course of antibiotics ended). A 29-week GA infant had bacteremia with no focus on day 39 and day 56 of life (7 days after a 10-day course of antibiotics ended). The third infant born at 27 weeks GA had LOD days 19 and day 36 of life (4 days after a 13-day course of antibiotics for bacteremia ended) with possible meningitis in the second episode (CSF pleocytosis in a culture-negative bloody tap obtained after antibiotics were given).

Fourteen of the 111 infants (13%) were discharged on anticonvulsants; 11 of these had confirmed and 2 had possible meningitis. Four infants had documented hearing loss, of whom 2 had meningitis. Three infants required ventriculoperitoneal shunts.

Five of 111 infants (5%) died before hospital discharge; 4 died on day 0–5 of treatment for GBS and one had withdrawal of care after GBS treatment was completed.

The number of cases over time is shown in Fig 1.

Serotypes from 100 of the 111 infants are shown in Fig 2 with 78 (78%) being serotype III (For the four infants with two episodes of LOD, serotypes were identified only on the initial isolate). Data were omitted from one infant as serotype III was identified in blood and serotype V in CSF; it seems likely that one of these results was incorrect, but specimens are no longer available for retesting.

### Very late onset disease (day 90–364)

There were 11 VLOD cases presenting on median day 116 (IQR 103–138) (range 93–207) of life (Table 1) (2 with positive blood and CSF cultures, 1 with positive CSF cultures only and 8 with positive blood and negative CSF cultures). Three infants were male (27%) and 8 (73%) were preterm. None had been diagnosed with chronic medical conditions at the time of GBS onset. Nine were bacteremic and two had only positive CSF cultures.

CSF was obtained from all 11 cases. Three (27%) were culture-positive (2 obtained prior to antibiotics and 1 after at least one dose antibiotics) and 8 (73%) were culture-negative (3 obtained prior to antibiotics, 5 after at least one dose antibiotics). None of the 5 with negative cultures obtained after antibiotics had CSF pleocytosis.

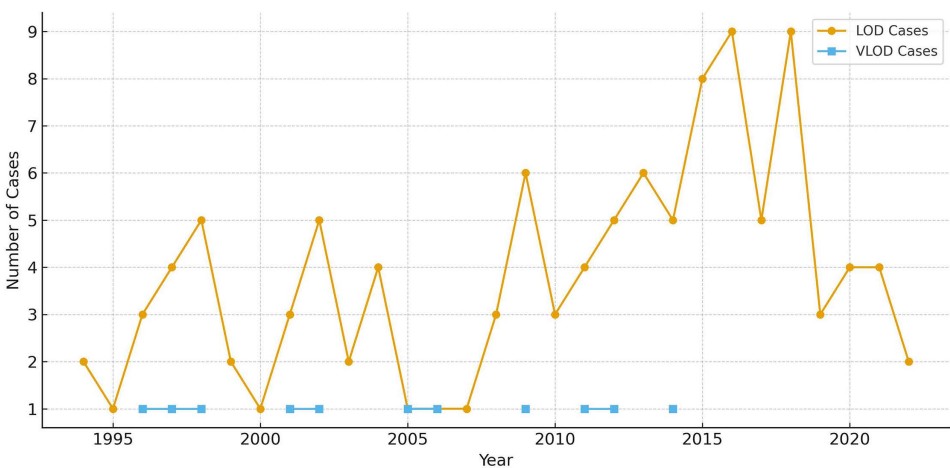

**Fig 1. Number of cases of GBS LOD and VLOD over time.**

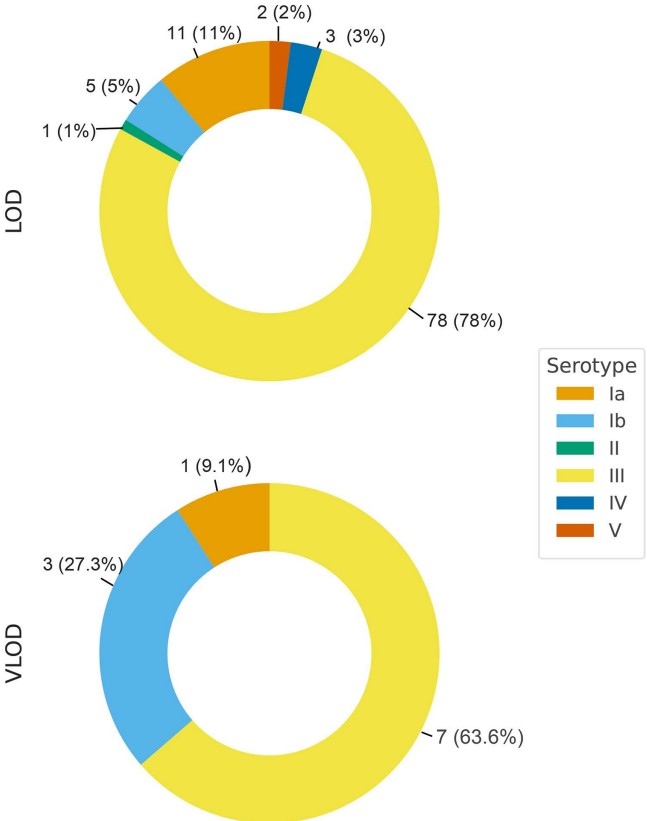

**Fig 2. GBS serotypes from LOD and VLOD cases N (%).**

Local sites of infection included 3 cases (27%) with cellulitis, none of whom had meningitis.

One of the 11 VLOD cases (9%) (with meningitis) was discharged on anticonvulsants. Six (55%) had normal hearing when tested while the other 5 (45%) were not tested. None had prior or subsequent GBS infections. All survived to discharge. Serotypes were Ia (N = 1; 9%)), Ib (N = 3; 27%), and III (N = 7; 64%) (Fig 2).

The number of cases overtime is shown in Fig 1 with there never being more than one VLOD case per year.

### Comparison of LOD to VLOD cases

As compared with infants with LOD, those with VLOD were less likely to be born vaginally (n = 3/11 [27%] versus n = 72/111 [65%] p = 0.036), more likely to be mechanically ventilated during their birth hospitalization (n = 5/11 [45%] versus n = 17/111 [15%]; p = 0.039); and more likely to have serious bacterial infections with other pathogens during their GBS admission (n = 3/11 [27%] versus n = 4/111 [4%]); p = 0.032) (Table 1). The proportion of isolates that were serotype III did not differ between LOD and VLOD cases (P = 0.51).

### Discussion

Over a 27-year period in Edmonton hospitals, there was a mean of 4.3 infants per year with GBS invasive disease with onset after day 6 of life (range 1–10 infants). Approximately 10% had VLOD (11 of 122). Statistically significant differences between LOD and VLOD were a higher incidence of vaginal delivery with LOD, and a higher incidence of mechanical

ventilation during birth hospitalization and serious bacterial infection with pathogens other than GBS noted with VLOD. This suggests that a risk factor for VLOD is having a complicated neonatal course.

From provincial data, the incidence of LOD in Alberta increased from 0.15 to 0.41 per 1000 live births from 2003 to 2020 for unclear reasons [9,10]. Data from the current study cannot be used to look at the incidence of LOD or VLOD as the proportion of infants with these conditions admitted to Edmonton hospitals is not known.

EOD always arises from intrapartum transmission due to high-grade GBS colonization. For LOD and VLOD, it is not clear when transmission of GBS most commonly occurs, i.e., intrapartum, from the mother after delivery (from GBS on her hands or from breastmilk), from nosocomial transmission, or from the community.

A 2022 systematic review of LOD reported an odds ratio (OR) of 2.67 (95% confidence interval (CI): 2.07–3.45) for maternal GBS colonization [13]. Furthermore, a prospective cohort of 100 infants with LOD reported that 30 of 47 mothers (64%) had GBS colonization [3]. In the current study, almost half of all infants with LOD where colonization status was known were born to those with colonization and almost half were preterm. In one study, IAP was associated with later presentation of LOD and milder disease [3]. This all suggests that intrapartum transmission accounts for some LOD.

Even less is known about the source of GBS or the characteristics of VLOD. Because of this, we completed a narrative review of the literature for VLOD case series. This identified 352 cases in 13 studies dating back to 1995 (Table 2) [14–26]. Three were population based studies that reported the incidence per 1000 live births: LOD 0.36 (95% CI: 0.31–0.42) and VLOD 0.03 (95% CI: 0.01–0.04) in New Zealand 2012–2021 [14], LOD 0.20 (95% CI: 0.17–0.23) and VLOD 0.012 (95% CI: 0.007–0.021) in Norway 1996–2012 [15], and LOD 0.12 (95% CI 0.11–0.14), and VLOD 0.01 (95% CI 0.01–0.02) in Japan 2011–2015 [16]. The upper age limit for VLOD was 6 months in the New Zealand study and 12 months in the other two studies. Combining data from the 7 studies that included cases up to 1 year of age [15–20. 22], there were 2242 cases of EOD, 2668 of LOD and 265 of VLOD – again indicating that as in our study, VLOD cases account for about 10% of GBS cases after 6 days of age. For the 10 studies that reported GA [14,17–26], 82 of 208 infants (39%) were preterm (versus 73% on the current study). Seven studies reported clinical presentation [15,17,18,22–24,26] with 64 of 187 infants (34%) having meningitis (versus 27% in the current study), 94 (54%) having bacteremia with no focus mentioned in the manuscript (versus 45% in the current study) and 20 (11%) having bacteremia with a likely focus: urinary tract infection (N = 7); cellulitis (n = 4); septic arthritis (N = 3); central line associated blood stream infection (N = 2); infective endocarditis (N = 2) and a bullous rash (N = 1).

In the literature review, a risk factor for VLOD as compared to LOD +/- EOD included prematurity in 4 studies [17,22,24,25] and male gender in 3 studies [17,20,22] (Table 2). However, combining data from the 6 studies that reported gender with our study [17,19,21–24,26], 95 of 176 VLOD cases were male (54%), so it seems unlikely that there is a true gender predominance. Another risk factor for VLOD in one study was the presence of underlying disorders (not further defined) [17]. A major limitation is that underlying disorders would often not yet be diagnosed at the time of LOD or VLOD presentation so are not reliably mentioned in published studies. Immune deficiency is thought to lead to some cases of VLOD. In the review of the literature, at least 2 of the 352 VLOD cases had human immunodeficiency virus and one study reported that 3 of 28 cases had "immunodeficiency". There is a case report of a child with IRAK-4 deficiency who presented with GBS meningitis at 5 months of age [28]. A recurrent case of VLOD has also been reported in a child with thoracic congenital lymphatic dysplasia (which is associated with CD4 lymphopenia and hypogammaglobulinemia) with two episodes of bacteremia 5 months apart starting at 2 years of age [29]. In our cohort, VLOD infants were more likely to have serious bacterial infections with pathogens other than GBS, which could signal a defect in host defense. However immune deficiency was not limited to VLOD cases, as one of our infants with LOD (GBS meningitis at 15 days of age) went on to be diagnosed with an Nf kappa B pathway immune deficiency that also affected toll-like receptor (TLR) function similar to IRAK-4 deficiency. The correlation between inborn errors of immunity that affect TLR function and VLOD/LOD GBS disease needs further study.

Table 2. Literature review of case series of VLOD due to GBS.

| Author Country Year | Study criteria | EOD[1] (N) LOD[2] (N) | Upper age for VLOD (years) | VLOD (N) | VLOD Male (%) | VLOD Preterm (%) | VLOD Bacteremia with no focus | VLOD Meningitis | Other presentation of VLOD | Underlying conditions apart from prematurity in VLOD cases | Age at onset of VLOD | Sequelae of VLOD | Comparison of VLOD with LOD |
|---|---|---|---|---|---|---|---|---|---|---|---|---|---|
| Millar New Zealand 2025 [14] | Infant in New Zealand with invasive GBS 2012–2021 | 190[3] 159 | 0.5 | 13 | NR | 9 (69%) | NR | NR | NR | NR | NR | NR | none |
| Mynarek Norway 2024 [15] | Infant in Norway 1996–2019 with invasive GBS (birth cohort=1,415,625)[4] | 501 341 | 1 | 24 | NR | NR | 15 (62%) | 9 (38%) | 0 | NR | NR | 5 died (21%) 7 NDD (37%) | Trend towards more deaths and NDD in VLOD cases than LOD or EOD (p value not calculated) |
| Sabroske US 2023 [16] | Infant in Houston 1970–2021 with invasive GBS | 945 976 | 1 | 96 | NR | NR | NR | NR | NR | NR | NR |  | none |
| Shibata Japan 2022 [17] | Surveillance questionnaire sent to 490 hospitals (343 responded) for invasive GBS cases 2016–2020 | 186 628 | 1 | 61 | 40 (66%) | 28 (46%) | 34 (56%) | 18 (30%) | N=9; UTI (N=7–11%); cellulitis (N=1–2%); septic arthritis (N=1–2%) | 8 (13%): CHD (N=4 including 2 with trisomy) and one each of hepatic failure, ileal perforation, megaureter and chondrodysplasia punctuata | median 119, IQR 100–149, range 90–312 | 2 died (One had meningitis) 7 had sequelae of which 2 were term with meningitis, 3 were preterm with meningitis, 2 were term with bacteremia and | VLOD cases were more likely to be male than LOD cases (66% versus 50%; p=0.022), more likely to be preterm (46% versus 25%; p=0.003), and more likely to have underlying conditions (13% versus 3%; p<0.001). |
| Mynarek Norway 2021 [18] | Infant in Norway 1996–2012 with invasive GBS (birth cohort=1,000,246) | 411 202 | 1 | 12 | NR | 5 (42%) | 6 (50%) | 6 (50%) | 0 | NR | NR | NR | none |
| Lee Taiwan 2019 [19] | Admitted to a single hospital with invasive GBS Jan 2013 – Jun 2017 | 7 55 | 1 | 7 | 6 (86%) | 1 (14%) | NR | 1 (14%) | NR | NR | median 140 (IQR 115–205) | 0 died | VLOD cases were more likely to be male than LOD cases but there were no differences in the incidence of prematurity, meningitis, complications or death |
| Kao Taiwan 2019 [20] | Admitted to a single hospital with invasive GBS 2003–2017 | 41 121 | 1 | 20 | NR | NR | NR | NR | NR | NR | NR | 2 (10%) died | none |

*(Continued)*

| Author Country Year | Study criteria | EOD[1] (N) LOD[2] (N) | Upper age for VLOD (years) | VLOD (N) | VLOD Male (%) | VLOD Preterm (%) | VLOD Bacteremia with no focus | VLOD Meningitis | Other presentation of VLOD | Underlying conditions apart from prematurity in VLOD cases | Age at onset of VLOD | Sequelae of VLOD | Comparison of VLOD with LOD |
|---|---|---|---|---|---|---|---|---|---|---|---|---|---|
| Zhang China 2019 [21] | Admitted to a single hospital with GBS meningitis Jan 2013-Oct 2017 | 18 71 | 16 | 7 | 5(71%) | 1 (14%) | Not applicable | 7 | Not applicable | NR | median 153 (range 95–214) | NR | VLOD cases were more likely to be male than LOD and EOD (13/18 vs. 28/71 vs. 5/7, $\chi^2$=7.705, P=0.024), and less likely to be admitted to ICU (5/18 vs. 31/71 vs. 0, $\chi^2$=6.065, P=0.042) |
| Matsubara Japan 2017 [22] | Surveillance questionnaire sent to 513 hospitals (250 responded) for invasive GBS cases 2011–2015 | 133 274 | 1 | 38[5] | 15 (39%) | 13 (34%) | 21 (55%) | 12 (32%) | N=5: Cellulitis (N=3–8%); septic arthritis (N=1–3%); UTI (N=1–3%) | 3 of 4 over 6 months of age had underlying conditions (liver transplant, hypogammaglobulinemia, and trisomy 21) | median 110 (IQR 99–147) range 91–319 | 0 died; 5.3% had sequelae (N=2- both term infants) | The median birth weight in EOD was significantly higher than in LOD (P<0.001) and VLOD (P=0.004). The proportion of low birth weight was increased with the age of onset (P for trend=0.008). The proportion with GA<37 weeks and GA<34 weeks was increased with age of onset (P=0.047 and 0.017, respectively) |

*(Continued)*

| Author Country Year | Study criteria | EOD[1] (N) LOD[2] (N) | Upper age for VLOD (years) | VLOD (N) | VLOD Male (%) | VLOD Preterm (%) | VLOD Bacteremia with no focus | VLOD Meningitis | Other presentation of VLOD | Underlying conditions apart from prematurity in VLOD cases | Age at onset of VLOD | Sequelae of VLOD | Comparison of VLOD with LOD |
|---|---|---|---|---|---|---|---|---|---|---|---|---|---|
| Bartlett Australia 2017 [23] | Admitted to one of two hospitals with invasive GBS 2000–2014 | NR 87 | 16 | 28 | 17 (61%) | 6/24 (25%) | 16 (57%) | 12 (43%) | 0 | 3 had immunodeficiency | median 118 (IQR 98–790) | 1 (4%) died; 5 (18%) had NDD | No difference in patient characteristics, presentation as meningitis versus bacteremia, ICU admission in LOD versus VLOD but LOD more likely to have sequelae defined as NDD or death than LOD (p=0.04; OR 3 [CI 1.03–8.77] (39% LOD vs 18% VLOD) |
| Cantey US 2014 [24] | Admitted to a single hospital with invasive GBS Jan 1, 2006-Jul 1 2012 | NR 42 (LOD starts day 4 of life) | 0.5 | 6 | 4 (67%) | 5 (83%) | 2 (33%) | 4 (67%) of whom one also had peritonitis | 0 | NR | median 102 days (IQR 95–110) | 0 died; 2 (33%) had CNS sequelae | No differences in clinical presentation, ICU admission, length of stay, NDD, or death when GA was controlled for. VLOD had lower median GA than LOD (28.5 vs. 39 weeks gestation, P<0.001). |
| Guilbert France 2010 [25][6] | Nationwide surveillance for GBS meningitis 2001–2006 | NR 220 | NR | 22 | NR | 9 (41%) | Not applicable | 22 | Not applicable | NR | mean 115 (range 91–226) | 1 (5%) died | No difference in complications (seizures. coma. shock, mechanical ventilation, death) or death alone in LOD versus VLOD but VLOD had significantly lower GA than LOD (35.6 weeks vs. 37.9 weeks, p=0.002). Prevalence of GA<32 weeks was higher in VLOD than LOD (32% vs. 7%, p=0.002 |

**Table 2.** (Continued)

| Author Country Year | Study criteria | EOD¹ (N) LOD² (N) | Upper age for VLOD (years) | VLOD (N) | VLOD Male (%) | VLOD Preterm (%) | VLOD Bacteremia with no focus | VLOD Meningitis | Other presentation of VLOD | Underlying conditions apart from prematurity in VLOD cases | Age at onset of VLOD | Sequelae of VLOD | Comparison of VLOD with LOD |
|---|---|---|---|---|---|---|---|---|---|---|---|---|---|
| Hussain US 1995 [26] | Admitted to a single hospital with invasive GBS Jan 1, 1986 – Jun 30, 1993 | 125 combined EOD and LOD | 18 | 18 – 9 were 15 weeks to one year) | 5 (28%) | 5 (28%) | 9 (50%) | 3 (25%) (all had hardware) | N = 6; Central venous catheter infection (N = 2); infective endocarditis (N = 2); septic arthritis (N = 1); bullous desquamation (N = 1) | N = 3 (33%) for those up to 1 year of age (2 with HIV and 1 with short gut syndrome); N = 6 (67%) for those 1–18 years of age (2 with CHD); 1 each of diabetes mellitus, Arnold Chiara malformation, ventriculoperitoneal shunt; AVM | NR | NR | none |

Legend: AVM – arteriovenous malformation; CHD – congenital heart disease; EOD – early onset disease; GA – gestational age; ICU – intensive care unit; LOD – late onset disease; NDD – neurodevelopmental disorder; US – United States; UTI – urinary tract infection; VLOD – very late onset disease.

¹defined as day 0–6 unless otherwise stated.

²day 7 to day 89.

³day 0–2 only – The number of EOD and LOD cases may not be accurate, as they are difficult to derive from the provided data.

⁴data from 866 of 1007 GBS cases where isolates were received by the Reference Laboratory.

⁵Two had prior LOD (both had meningitis the first episode and bacteremia with no focus the second episode) and one had a second episode of VLOD 21 days after the first episode (7 days after antibiotics stopped) with bacteremia with no focus both episodes.

⁶appears to overlap with patients reported in Georget-Bouquin study [27] also identified by our search so only data from this study included in table.

In the literature review, in terms of severity of illness, one study reported at least a trend towards a higher risk of neuro-developmental disorders (NDD) and of death with VLOD than with EOD or LOD (P value not provided) [15] while another study reported the opposite – a decreased risk of NDD or death with VLOD than with LOD [23] (Table 2). In the current study, there were no deaths in 11 VLOD cases versus 5 deaths in 111 infants with LOD; neurodevelopmental outcomes were not captured. One previous study reported a lower risk of ICU admission with VLOD than with LOD and EOD combined [21].

It is possible that the pathogenicity of certain serotypes of GBS predisposes to later onset disease. Serotype III is the most common cause of both EOD and LOD infant invasive GBS in the literature [30] and in the current study. Sequence type 17 and its associated virulence factor HvGA are associated with infant invasive GBS, with the majority of these cases being serotype III [30]; unfortunately, such testing was not available for the isolates from the current study.

Serotyping was reported for 180 of the 352 VLOD cases in the literature review [16,17,19,20,25,26] including 87 (48%) serotype III, 44 (24%) serotype Ia, 17 (9%) serotype Ib, 13 (7%) serotype V, 10 (6%) serotype II, 5 (3%) serotype VI, 3 (2%) serotype IV and 1 (1%) serotype IX. In in our cohort, serotype III was also the most common accounting for 64% of VLOD cases ((Fig 2). In the literature review, some VLOD cases were with uncommon serotypes (VI: N = 5; XI: N = 1) and there is a case report of VLOD meningitis due to serotype IX presenting at 120 days of age [31]. This may further support the hypothesis that a defect in host defense is sometimes the culprit in VLOD, as less pathogenic serotypes of GBS are sometimes responsible for VLOD cases.

It has been postulated that VLOD has the same origins and clinical course as LOD, occurring in infants who were typically more preterm so present at an older chronological age [24]. However, this does not explain VLOD cases in term babies, which accounted for 61% of cases in the literature review (Table 2) and about one-third of cases in the current study. Our review of the VLOD literature suggests that the risk factors, clinical course, serotypes and outcomes are broadly similar to those of LOD.

Limitations of the study are all those typical of a retrospective chart review. Serotyping data was incomplete.

In conclusion, our study shows that VLOD accounts for about 10% of GBS disease after 6 days of age, but there are only relatively small studies looking at this important cause of morbidity and mortality. While prematurity is a risk factor in previous studies, further exploration into defects in infant microbial defense, particularly innate immune defects, is warranted. It seems possible that infant GBS shifts from being an expected pathogen to an opportunistic one with maturity and that an immune disorder should be considered in infants with VLOD.

## Supporting information

**S1 Table. Discharge screening codes.** ICD9 CM and ICD19 CA Discharge Codes used for health records retrieval. (DOCX)

**S2 Table. Case report form.** REDCap form used for data collection. (PDF)

## Author contributions

**Conceptualization:** Sneha Suresh.

**Data curation:** Isoken Isah, Gregory J. Tyrrell.

**Formal analysis:** Manoj Kumar.

**Methodology:** Joan L. Robinson, Sneha Suresh, Gregory J. Tyrrell.

**Supervision:** Joan L. Robinson, Sneha Suresh.

**Visualization:** Sneha Suresh.

**Writing – original draft:** Joan L. Robinson.

**Writing – review & editing:** Sneha Suresh, Manoj Kumar, Isoken Isah, Gregory J. Tyrrell.

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
