## [Decision Letter · Decision Letter 0]

1 Dec 2025

Dear Dr.  Robinson,

Thank you for submitting your manuscript to PLOS ONE. After careful consideration, we feel that it has merit but does not fully meet PLOS ONE’s publication criteria as it currently stands. Therefore, we invite you to submit a revised version of the manuscript that addresses the points raised during the review process.

We look forward to receiving your revised manuscript.

Kind regards,

Yung-Fu Chang

Academic Editor

PLOS ONE

Journal Requirements:

3. Please include a caption for figure 1.

Reviewers' comments:

Reviewer's Responses to Questions

**Comments to the Author**

1. Is the manuscript technically sound, and do the data support the conclusions?

Reviewer #1: Partly

Reviewer #2: Partly

2. Has the statistical analysis been performed appropriately and rigorously?

Reviewer #1: Yes

Reviewer #2: Yes

3. Have the authors made all data underlying the findings in their manuscript fully available?

Reviewer #1: No

Reviewer #2: Yes

4. Is the manuscript presented in an intelligible fashion and written in standard English?

Reviewer #1: Yes

Reviewer #2: Yes

Reviewer #1: The authors conducted a retrospective analysis of the clinical and microbiological characteristics of late-onset disease (LOD) and very late-onset disease (VLOD) caused by group B streptococcus (GBS) over a 27-year period at four hospitals in Edmonton. This long-term epidemiological study contains valuable information; however, the following points require clarification or revision.

Major Issues

Comment 1

The stated study objective—“The objective of this study was to describe and compare the incidence and characteristics of LOD and VLOD in a cohort of neonates in a Canadian city over a prolonged period.”—is not adequately reflected in the title or the conclusions of the abstract. Revision is required to ensure consistency with the study objective.

Comment 2

Figure 1 is neither described nor cited anywhere in the main text. Please add an explanation of the figure and cite it appropriately in the manuscript.

Comment 3

Page 5, Lines 79–83

“The GBS prenatal protocol during these years included rectovaginal screening at 35 to 37 weeks gestation with intrapartum antibiotics for all who screened positive and for unscreened persons with risk factors for GBS transmission.”

>>Were national or provincial guidelines for GBS prevention available in Canada during the 1990s? Additionally, were such guidelines followed at the participating institutions? The following guidelines were published in 1997 and 2002, but please describe briefly—citing appropriate literature—the GBS screening practices and intrapartum prophylaxis protocols used in Canada and at your institutions before these guidelines were issued:

• Society of Obstetricians and Gynaecologists. Statement on the prevention of early-onset group B streptococcal infections in the newborn. J Soc Obstet Gynaecol. 1997;19:751–8.

• Prevention of group B streptococcal infection in newborns: recommendation statement from the Canadian Task Force on Preventive Health Care. CMAJ. 2002;166:928–30.

Comment 4

Page 6, Lines 103–105

Incidence was calculated using Statista live-birth data for Alberta; however, using population-based provincial data as the denominator requires justification. Please clarify the relationship between births at the four study hospitals and the overall number of births in Edmonton and in Alberta. Specifically, indicate what proportion of Edmonton/Alberta births occurred at the study sites and whether these proportions remained stable over the 27-year study period.

Comment 5

Discussion

Each paragraph summarizes findings from previous literature, but the discussion lacks deeper comparison with, and interpretation in the context of, the present study’s results. Please revise to highlight similarities, differences, and the implications of your findings.

Comment 6

Please clarify whether the study population overlaps with previously published cohorts cited as references 2 and 6. If so, this should be explicitly stated in the Methods section, including a brief note confirming that the present manuscript does not constitute a duplication or overlap report.

• Ref 2: Ma A et al., Microbiol Spectr. 2021.

• Ref 6: Alhhazmi A et al., J Clin Microbiol. 2016.

Minor Issues

Comment 1

Page 5, Lines 91–92

The definition of CSF pleocytosis differs for infants <1 month and ≥1 month of age. Please provide the age-appropriate thresholds. The value “> 20 × 10⁹/L WBCs” is clearly incorrect and should be corrected.

Additionally, the ratio “1000 RBCs : 1 WBC” appears to indicate correction for traumatic taps; please state this explicitly and revise for accuracy.

Comment 2

Table 1

Please add a row showing the gestational age (median and IQR) for preterm infants only.

Comment 3

Page 10, Lines 179–180

The text states:

“Local sites of infection included 3 cases with cellulitis, none of whom had meningitis. One infant (with meningitis) was discharged on anticonvulsants.”

These two sentences contradict each other. Please revise for consistency.

Comment 4

Figure 2

Please clarify what “historic VLOD cases reported in the literature” refers to. Why is historical data shown only for VLOD? The origin of these data should be explained in the caption or the text.

Comment 5

Pages 18 and 19

Line numbering is missing. Please correct.

Comment 6

Page 18

“However, combining data from the 6 studies that reported gender with our study…”

Please identify these six studies.

Comment 7

Page 19

“Serotyping was reported for 180 of the 352 VLOD cases in the literature review…”

Please add the appropriate references.

Comment 8

Page 19

“In the literature review, some VLOD cases were with t uncommon serotypes (VI: N=5; XI: N=1)”

The phrase “t uncommon serotypes” appears to be a typographical error. Please correct.

Comment 9

Figure 1

Please use a linear (not logarithmic) X-axis. Also adjust font size to ensure legibility in print.

Reviewer #2: This manuscript presents a long-term cohort study from a Canadian city describing late-onset disease (LOD) and very-late-onset disease (VLOD) due to Group B Streptococcus (GBS). The paper is well written, clearly presented, and supported by a transparent methodology with appropriately analysed results. The inclusion of a literature review of VLOD adds value and situates the findings within the broader epidemiologic context. Overall, the study offers useful descriptive data.

Major Comments

Discussion and conclusions need closer alignment with study data.

The main weakness is that the discussion is somewhat diffuse and not consistently tied to the results. The conclusions would benefit from clearer, data-driven statements. A focused rewrite of the discussion section is recommended, structured around the following key points:

Incidence trends:

The manuscript suggests—or may imply—an increase in LOD incidence over the study period, but this is not clearly articulated or quantified. If incidence has increased, this should be explicitly stated, contextualised with regional/national surveillance data, and compared with international literature.

In addition, the absence of whole-genome sequencing limits the ability to comment on changes in strain invasiveness or the emergence of more virulent lineages; this constraint should be acknowledged as it affects interpretation of temporal trends.

Risk factors for LOD:

Although not clearly stated, prematurity and maternal colonisation appear to be risk factors for LOD in this cohort when compare to the general population. These findings align well with existing literature. This point should be highlighted more directly as an area of agreement with previous studies.

Routes of transmission for LOD:

The current discussion devotes considerable space to possible transmission pathways for LOD. As the dataset does not enable meaningful conclusions about specific routes, this section should be shortened. A concise statement acknowledging the uncertainty is sufficient; extensive speculation is unnecessary and distracts from the strengths of the study.

VLOD proportion:

Approximately 10% of cases occurred after day 90, which is consistent with the provided literature review. This is an important and often overlooked clinical entity. The authors should emphasise its significance: VLOD is likely under-recognised in routine surveillance, and the findings support the need for improved clinical awareness and further research into its epidemiology and immunologic determinants.

Comparing LOD and VLOD:

The study reports several differences between LOD and VLOD, but the VLOD sample is small and many observed differences are not clinically meaningful. Stronger conclusions can be drawn when these data are interpreted alongside the literature review. Specifically:

a. Risk factors such as prematurity and maternal colonisation appear broadly similar.

b. The possibility of “immunity gaps” being more common in VLOD is intriguing and should be framed cautiously as a hypothesis supported by literature, rather than a definitive finding.

c. Clinical presentation—including high rates of meningitis—is similar in both groups.

d. Outcomes do not substantially differ.

e. Serotype distribution is similar.

The final paragraph of the discussion already summarises this well; this could serve as the central message and the rest of the discussion streamlined to support it.

Minor Comments

1. Lines 49–51: Please report the absolute numbers, percentages, and corresponding p-values for clarity.

2. Lines 139–140: The definitions of “confirmed” and “possible” meningitis should be explicitly stated in the Methods section.

3. Lines 187–189: As above, please include numbers, percentages, and p-values to allow clear interpretation.

4. Lines 200–203: This appears to describe a VLOD case; consider removing this text to avoid confusion.

5. Lines 210–213: Routes of transmission may overlap between EOD and LOD, although the pathogenesis differs (e.g., persistent colonisation in LOD vs acute high-inoculum infection in EOD). This distinction is well established in the literature and could be clarified accordingly.

6. Lines 214–225: I recommend removing this section, consistent with major comments regarding limiting speculation on transmission routes.

7. Lines 228–232: Numbered lists (i, ii, iii, etc.) are not typically used in scientific manuscripts; consider integrating these points into the narrative text.

8. Figure 1: The incidence is unclear. Please revise the y-axis labelling so that the incidence rate is immediately interpretable.

9. Figure 2: Consider adding numbers and/or percentages to the ring chart to enhance readability.

**Do you want your identity to be public for this peer review?** For information about this choice, including consent withdrawal, please see our Privacy Policy

Reviewer #1: No

Reviewer #2: No

---

## [Author Response · Author response to Decision Letter 1]

15 Dec 2025

Response: Done

Response: We asked our ethics committee and they informed us that we did not have permission to place the data in a repository. The letter from them stating that is now uploaded as a supporting document. Inquiries regarding these restrictions should be directed to Charmaine Kabatoff (kabatoff@ualberta.ca).

3. Please include a caption for figure 1.

Response: Done

Response: Done

Response: Reviewer comments did not suggest quoting specific studies.

Reviewers' comments:

Reviewer's Responses to Questions

Comments to the Author

Reviewer #1: The authors conducted a retrospective analysis of the clinical and microbiological characteristics of late-onset disease (LOD) and very late-onset disease (VLOD) caused by group B streptococcus (GBS) over a 27-year period at four hospitals in Edmonton. This long-term epidemiological study contains valuable information; however, the following points require clarification or revision.

Major Issues

Comment 1

The stated study objective—“The objective of this study was to describe and compare the incidence and characteristics of LOD and VLOD in a cohort of neonates in a Canadian city over a prolonged period.”—is not adequately reflected in the title or the conclusions of the abstract. Revision is required to ensure consistency with the study objective.

Response: Thanks for pointing out this deficiency. The title was changed to “Shifting From an Expected to an Opportunistic Pathogen: Comparison of Cases of Infant Late and Very Late Onset Group B streptococcal (GBS) infection in a Canadian city over a 27-year period”. The conclusion of the abstract now states: ‘Infants with VLOD appear to have more complex medical histories prior to GBS infection than do those with LOD.”

Comment 2

Figure 1 is neither described nor cited anywhere in the main text. Please add an explanation of the figure and cite it appropriately in the manuscript.

Response: Done

Comment 3

Page 5, Lines 79–83

“The GBS prenatal protocol during these years included rectovaginal screening at 35 to 37 weeks gestation with intrapartum antibiotics for all who screened positive and for unscreened persons with risk factors for GBS transmission.”

>>Were national or provincial guidelines for GBS prevention available in Canada during the 1990s? Additionally, were such guidelines followed at the participating institutions? The following guidelines were published in 1997 and 2002, but please describe briefly—citing appropriate literature—the GBS screening practices and intrapartum prophylaxis protocols used in Canada and at your institutions before these guidelines were issued:

• Society of Obstetricians and Gynaecologists. Statement on the prevention of early-onset group B streptococcal infections in the newborn. J Soc Obstet Gynaecol. 1997;19:751–8.

• Prevention of group B streptococcal infection in newborns: recommendation statement from the Canadian Task Force on Preventive Health Care. CMAJ. 2002;166:928–30.

Response: Thanks for pointing out that we had forgotten that routine screening for GBS colonization was not in place in the early years of this study. It is difficult to know how quickly clinicians adopted the recommendations of the SOGC or of the Canadian Task Force on Preventive Health Care. We therefore state: “The GBS prenatal protocol during most of these years included rectovaginal screening at 35 to 37 weeks gestation with intrapartum antibiotics for all who screened positive and for unscreened persons with risk factors for GBS transmission. In the early years of the study, clinicians were just becoming aware of the benefits of GBS screening, so some did not screen or gave intrapartum antibiotics only in the presence of risk factors for GBS transmission.”

Comment 4

Page 6, Lines 103–105

Incidence was calculated using Statista live-birth data for Alberta; however, using population-based provincial data as the denominator requires justification. Please clarify the relationship between births at the four study hospitals and the overall number of births in Edmonton and in Alberta. Specifically, indicate what proportion of Edmonton/Alberta births occurred at the study sites and whether these proportions remained stable over the 27-year study period.

Response: The problem is that we enrolled infants admitted to one of the four Edmonton hospitals, some of whom were born outside of Edmonton. Therefore there is not a reliable denominator so we deleted this part of the data.

Comment 5

Discussion

Each paragraph summarizes findings from previous literature, but the discussion lacks deeper comparison with, and interpretation in the context of, the present study’s results. Please revise to highlight similarities, differences, and the implications of your findings.

Response: Throughout the discussion, we added data from the current study for comparison where appropriate.

Comment 6

Please clarify whether the study population overlaps with previously published cohorts cited as references 2 and 6. If so, this should be explicitly stated in the Methods section, including a brief note confirming that the present manuscript does not constitute a duplication or overlap report.

• Ref 2: Ma A et al., Microbiol Spectr. 2021.

• Ref 6: Alhhazmi A et al., J Clin Microbiol. 2016.

Response: In the Methods section, we now state: “For the 2003-2020 cases, microbiologic data only was included in two previous provincial studies [2,3].“

Minor Issues

Comment 1

Page 5, Lines 91–92

The definition of CSF pleocytosis differs for infants <1 month and ≥1 month of age. Please provide the age-appropriate thresholds. The value “> 20 × 10⁹/L WBCs” is clearly incorrect and should be corrected.

Additionally, the ratio “1000 RBCs : 1 WBC” appears to indicate correction for traumatic taps; please state this explicitly and revise for accuracy.

Response: We now state:” Cerebrospinal fluid (CSF) pleocytosis was defined as > 20 X 109/L WBCs based on the assumption that many infants would be preterm. For traumatic taps, a ratio of 1000 RBCs:1WBC was applied. “ It is difficult to know the optimal cut-off for CSF pleocytosis in a preterm infant over 30 days of age which is why we selected a cut-off of 20 × 10⁹/L. The reviewer is correct that a lower number would be appropriate in term infants over 30 days of age. Looking at our data, there were 3 infants with 6 to 20 × 10⁹/L WBC’s after 30 days of age; i) term infant with 6 WBCs at 39 days of age, ii) infant born at 29 weeks gestation with 9 WBCs at 40 days of age and iii) infant born at 32 weeks gestation with 9 WBCs at 50 days of age. It does not seem very likely that these infants had bacterial meningitis. However, if the reviewer disagrees, we can change the methods so these infants are counted as having possible GBS meningitis.

Comment 2

Table 1

Please add a row showing the gestational age (median and IQR) for preterm infants only.

Response: Done

Comment 3

Page 10, Lines 179–180

The text states:

“Local sites of infection included 3 cases with cellulitis, none of whom had meningitis. One infant (with meningitis) was discharged on anticonvulsants.”

These two sentences contradict each other. Please revise for consistency.

Response: Thanks for pointing out that this could be confusing. We now try to make it clearer that the sentence about the anticonvulsants applies to all VLOD cases, not the 3 with meningitis. We now state: “Local sites of infection included 3 cases with cellulitis, none of whom had meningitis. One of the 11 VLOD cases (with meningitis) was discharged on anticonvulsants.”

Comment 4

Figure 2

Please clarify what “historic VLOD cases reported in the literature” refers to. Why is historical data shown only for VLOD? The origin of these data should be explained in the caption or the text.

Response: We agree that this is confusing. The caption was changed to “GBS serotypes from LOD and VLOD cases in the current study and previous VLOD cases reported in the literature”.

We reviewed VLOD cases but not LOD cases in the literature for this manuscript. It would be a daunting task to review previously reported LOD cases.

Comment 5

Pages 18 and 19

Line numbering is missing. Please correct.

Response: This was corrected.

Comment 6

Page 18

“However, combining data from the 6 studies that reported gender with our study…”

Please identify these six studies.

Response: Done

Comment 7

Page 19

“Serotyping was reported for 180 of the 352 VLOD cases in the literature review…”

Please add the appropriate references.

Response: Done

Comment 8

Page 19

“In the literature review, some VLOD cases were with t uncommon serotypes (VI: N=5; XI: N=1)”

The phrase “t uncommon serotypes” appears to be a typographical error. Please correct.

Response: This was corrected

Comment 9

Figure 1

Please use a linear (not logarithmic) X-axis. Also adjust font size to ensure legibility in print.

Response: This figure was changed so there is no longer a logarithmic axis.

Reviewer #2: This manuscript presents a long-term cohort study from a Canadian city describing late-onset disease (LOD) and very-late-onset disease (VLOD) due to Group B Streptococcus (GBS). The paper is well written, clearly presented, and supported by a transparent methodology with appropriately analysed results. The inclusion of a literature review of VLOD adds value and situates the findings within the broader epidemiologic context. Overall, the study offers useful descriptive data.

Major Comments

Discussion and conclusions need closer alignment with study data.

The main weakness is that the discussion is somewhat diffuse and not consistently tied to the results. The conclusions would benefit from clearer, data-driven statements. A focused rewrite of the discussion section is recommended, structured around the following key points:

Incidence trends:

The manuscript suggests—or may imply—an increase in LOD incidence over the study period, but this is not clearly articulated or quantified. If incidence has increased, this should be explicitly stated, contextualised with regional/national surveillance data, and compared with international literature.

In addition, the absence of whole-genome sequencing limits the ability to comment on changes in strain invasiveness or the emergence of more virulent lineages; this constraint should be acknowledged as it affects interpretation of temporal trends.

Response: Unfortunately, our methodology precludes us from reaching any definitive conclusion about the incidence of LOD as we looked only at infants admitted to four hospitals in Edmonton and referral patterns change over time. Therefore, there is not a reliable denominator so we deleted incidence data from Fig 1. We now state in the discussion:” From provincial data, the incidence of LOD in Alberta increased from 0.15 to 0.41 per 1000 live births from 2003 to 2020 for unclear reasons [2,3]. Data from the current study cannot be used to look at the incidence of LOD or VLOD as the proportion of infants with these conditions admitted to Edmonton hospitals is not known. “

Risk factors for LOD:

Although not clearly stated, prematurity and maternal colonisation appear to be risk factors for LOD in this cohort when compare to the general population. These findings align well with existing literature. This point should be highlighted more directly as an area of agreement with previous studies.

Response: In the discussion (line 247) we now state:” [7]. In the current study, almost half of all infants where colonization status was known were born to those with colonization and almost half were preterm.”

Routes of transmission for LOD:

The current discussion devotes considerable space to possible transmission pathways for LOD. As the dataset does not enable meaningful conclusions about specific routes, this section should be shortened. A concise statement acknowledging the uncertainty is sufficient; extensive speculation is unnecessary and distracts from the strengths of the study.

Response: This section was deleted.

VLOD proportion:

Approximately 10% of cases occurred after day 90, which is consistent with the provided literature review. This is an important and often overlooked clinical entity. The authors should emphasise its significance: VLOD is likely under-recognised in routine surveillance, and the findings support the need for improved clinical awareness and further research into its epidemiology and immunologic determinants.

Response: Our conclusion now starts with “Our study shows that VLOD accounts for about 10% of GBS disease after 6 days of age, but there are only relatively small studies looking at this important cause of morbidity and mortality. “

Comparing LOD and VLOD:

The study reports several differences between LOD and VLOD, but the VLOD sample is small and many observed differences are not clinically meaningful. Stronger conclusions can be drawn when these data are interpreted alongside the literature review. Specifically:

a. Risk factors such as prematurity and maternal colonisation appear broadly similar.

b. The possibility of “immunity gaps” being more common in VLOD is intriguing and should be framed cautiously as a hypothesis supported by literature, rather than a definitive finding.

c. Clinical presentation—including high rates of meningitis—is similar in both groups.

d. Outcomes do not substantially differ.

e. Serotype distribution is similar.

The final paragraph of the discussion already summarises this well; this could serve as the central message and the rest of the discussion streamlined to support it.

Response: We have tried to improve the discussion as suggested. O

---

## [Decision Letter · Decision Letter 1]

6 Jan 2026

Dear Dr. Robinson,

Thank you for submitting your manuscript to PLOS ONE. After careful consideration, we feel that it has merit but does not fully meet PLOS ONE’s publication criteria as it currently stands. Therefore, we invite you to submit a revised version of the manuscript that addresses the points raised during the review process.

plosone@plos.org . A letter that responds to each point raised by the academic editor and reviewer(s). You should upload this letter as a separate file labeled 'Response to Reviewers'.A marked-up copy of your manuscript that highlights changes made to the original version. You should upload this as a separate file labeled 'Revised Manuscript with Track Changes'.An unmarked version of your revised paper without tracked changes. You should upload this as a separate file labeled 'Manuscript'.

We look forward to receiving your revised manuscript.

Kind regards,

Yung-Fu Chang

Academic Editor

PLOS One

Journal Requirements:

Reviewers' comments:

Reviewer's Responses to Questions

**Comments to the Author**

Reviewer #1: All comments have been addressed

Reviewer #2: (No Response)

2. Is the manuscript technically sound, and do the data support the conclusions?

Reviewer #1: Partly

Reviewer #2: Yes

3. Has the statistical analysis been performed appropriately and rigorously?

Reviewer #1: No

Reviewer #2: Yes

4. Have the authors made all data underlying the findings in their manuscript fully available?

Reviewer #1: Yes

Reviewer #2: Yes

5. Is the manuscript presented in an intelligible fashion and written in standard English?

Reviewer #1: Yes

Reviewer #2: Yes

Reviewer #1: The authors conducted a retrospective analysis of the clinical and microbiological characteristics of late-onset disease (LOD) and very late-onset disease (VLOD) caused by group B streptococcus (GBS) over a 27-year period at four hospitals in Edmonton. While the revised manuscript has improved, the following points require clarification or further revision.

Major comments

Comment 1

The authors stated “The GBS prenatal protocol during most of these years included rectovaginal screening at 35 to 37 weeks gestation with intrapartum antibiotics for all who screened positive and for unscreened persons with risk factors for GBS transmission. In the early years of the study, clinicians were just becoming aware of the benefits of GBS screening, so some did not screen or gave intrapartum antibiotics only in the presence of risk factors for GBS transmission.”

As I indicated in my comments on the original manuscript, the authors should more clearly explain when the Canadian guidelines regarding rectovaginal culture screening and intrapartum antibiotic prophylaxis for the prevention of early-onset GBS disease were implemented during the study period. Relevant guideline documents should be cited to clarify changes in clinical practice over time. The authors’ explanation of the revised manuscript is ambiguous for readers.

Comment 2

As I indicated in my comments on the original manuscript, the definition of CSF pleocytosis differs for infants <1 month and ≥1 month of age.

The authors’ response stating that “cerebrospinal fluid (CSF) pleocytosis was defined as >20 × 10⁹/L white blood cells based on the assumption that many infants would be preterm” is not scientifically justified. In cases of late-onset disease (LOD), the proportion of preterm births is less than half, and therefore this assumption does not adequately support the chosen threshold.

The authors should define the criterion used for CSF pleocytosis and revise the number of cases classified as possible meningitis accordingly. As a consequence, the conclusions of the manuscript may also require revision.

The authors are encouraged to review relevant studies, including but not limited to the following, and to cite literature that best supports the CSF reference values adopted in the present study:

- Zimmermann P, et al. Normal Values for Cerebrospinal Fluid in Neonates: A Systematic Review. Neonatology. 2021.

- Kestenbaum LA, et al. Defining cerebrospinal fluid white blood cell count reference values in neonates and young infants. Pediatrics. 2010.

Comment 3

The unit used for the CSF WBC count appears to be incorrect. CSF WBC counts are expressed as ×10⁶/L (equivalent to cells/µL). Accordingly, the statement defining CSF pleocytosis as “>20 ×10⁹/L WBCs” is a critical error. This is not merely a semantic issue, as the definition of pleocytosis directly affects case classification.

Comment 4

In response to my comment that the original manuscript should clarify what historic VLOD cases reported in the literature refers to, the authors state that they reviewed VLOD cases but not LOD cases in the literature. They note that reviewing previously reported LOD cases would be a daunting task. While this explanation is understandable from a practical standpoint, it does not fully align with the stated objective of the study.

The authors describe their objectives as “to describe and compare the incidence and characteristics of LOD and VLOD in a cohort of neonates in a Canadian city over a prolonged period”. Based on this objective, it is unclear why the literature review focuses exclusively on VLOD, and why previously reported data are not addressed in the same way for LOD.

If the authors believe that a focused review of VLOD cases is particularly warranted, this rationale should be stated in the manuscript. Specifically, an explicit explanation should be added before the section reviewing VLOD cases in the Discussion to clarify why only VLOD is reviewed and how this approach supports the overall aims of the study.

Comment 5

Related to Comment 4, the presentation of serotype distribution in Figure 2 appears incomplete. Showing only previously reported VLOD cases, without corresponding data for LOD, provides an unbalanced view and does not fully align with the overall study objective.

I therefore suggest removing the previously reported VLOD data from Figure 2. Instead, a brief comparison of serotype distributions between the present study and previously reported VLOD cases could be more appropriately addressed in the Discussion.

Comment 6 (Statistical analysis)

In the Abstract, main text, and tables, the authors report comparisons such as “n = 3 (27%) versus n = 72 (65%); p = 0.036” and “n = 5 (45%) versus n = 17 (15%); p = 0.039.” Based on these percentages, it appears that the denominators may be 11 and 111, respectively; however, this is not explicitly stated and should be clarified.

In the Methods section, the authors indicate that Fisher’s exact test was used when the expected cell frequency was <5 in any cell of a 2×2 table. Therefore, it is assumed that Fisher’s exact test was applied to these comparisons. However, when I applied Fisher’s exact test using my standard statistical software, however, the reported p values could not be reproduced.

To ensure transparency and reproducibility, the authors are requested to provide the underlying 2×2 contingency tables and the corresponding results of Fisher’s exact test for these comparisons. It would be sufficient to provide this information in the response to the reviewers. Concomitantly, other statistical analyses are required to be re-examined.

Comment 7 (Introduction)

In the Introduction, only reference #1 is cited. This is somewhat unexpected, as the Introduction is intended to summarize both what is already known and what remains unknown in the field. To provide appropriate context for the study, it would be helpful to include additional key references that reflect the existing body of literature.

Incorporating multiple relevant citations would strengthen the background section and better support the rationale for the present study.

Comment 8 (Table 1)

The footnotes to Table 1, under “Invasive infection with bacteria other than GBS during GBS admission,” list the entries

“5 Urinary tract infection (N = 1)” and “6 Urinary tract infection (N = 1)”.

However, a urinary tract infection is generally not typically classified as an invasive bacterial infection. Therefore, these entries should therefore be removed from this category, and the corresponding statistical analyses should be revised. Any affected results should also be corrected in the Abstract and the main text.

Comment 9 (Table 1)

Table 1 lists “5 coagulase-negative staphylococci bacteremia (N = 3)” and “6 coagulase-negative staphylococci bacteremia (N = 1), meningitis due to Ureaplasma urealyticus.” Coagulase-negative staphylococcal bacteremia is unlikely represent a true invasive infection occurring concurrently with GBS bacteremia; these findings are more consistent with contamination.

In addition, the entry “coagulase-negative staphylococci bacteremia (N = 1), meningitis due to Ureaplasma urealyticus” appears to list two different pathogens. It is unclear whether this case is being classified as having three invasive infections (GBS bacteremia, Ureaplasma urealyticus meningitis, and coagulase-negative staphylococcal bacteremia).

The authors should clarify how these cases were adjudicated, to justify the classification used, and to revise Table 1 and any related analyses if necessary.

Minor comments

Comment 1 (Line 123)

“Screening of discharge codes yielded 842 infants of which 122 had confirmed LOS.”

Please clarify what “LOS” refers to, as it is not defined at this point in the manuscript.

Comment 2 (Table 1)

Although 111 infants are listed under LOD, there are 4 recurrent cases, resulting in a total of 115 episodes. The following variables appear to be episode-based and should therefore be analyzed on an episode basis (115 episodes) rather than by individual infants (N = 111):

– Mechanical ventilation during birth hospitalization

– NEC

– Invasive infection with bacteria other than GBS during GBS admission

– Day of onset

Comment 3 (Table 1)

“Mechanical ventilation during birth hospitalization”

Consider simplifying this to “Mechanical ventilation.”

Comment 4 (Table 1)

Please add the proportions of bacteremia, proven meningitis, and possible meningitis to Table 1, based on the criteria described in Major Comment 2.

Comment 5 (Figure 2)

In lines 175–176, the authors state that serotypes from 101 of the 111 infants with LOD are shown in Figure 2, with 78 (77%) being serotype III (noting that for four infants with two episodes of LOD, serotypes were identified only from the initial isolate). However, the numbers and percentages presented in Figure 2 do not appear to match these values. Please reconcile the data in the text and Figure 2.

Comment 6 (Line 180)

“Figure 1 – Incidence of GBS LOD and VLOD over time”

The authors appear to have removed the incidence from figure. Please confirm and revise this caption accordingly.

Comment 7 (Lines 191–192)

The manuscript states: “None of the 5 with negative cultures obtained after antibiotics had CSF pleocytosis.” However, in the LOD analysis, such cases were classified as possible meningitis.

If these five cases are interpreted as possible meningitis, this classification should be explicitly stated to ensure consistency in case definitions and interpretation. The authors are requested to clarify how these cases were categorized and to revise the text and Table accordingly if necessary.

Comment 8 (Lines 197–198)

“Serotypes were Ia (N = 1; 9%)), Ib (N = 3; 27%))”

There is a duplication of closing parentheses that should be corrected.

Comment 9 (Lines 221–222)

The discussion states that, for LOD, it is unclear when transmission of GBS most commonly occurs (e.g., intrapartum, postnatal maternal transmission, nosocomial, or community sources). This consideration also applies to VLOD. Given that the manuscript addresses both LOD and VLOD, it would be preferable to discuss these transmission routes for both entities.

Comment 10 (Table 2)

“Kao Taiwan 2019([14])”

Please revise to “Kao Taiwan 2019 [14]” for consistency.

Comment 11 (Line 241)

“Combining data from the 8 studies that included cases up to 1 year of age”

Please specify which eight studies are being referred to.

Comment 12 (Line 244)

“For the 10 studies that reported GA, 82 of 208 infants (39%) were preterm (versus 73% in the current study).”

References supporting this statement should be provided.

Comment 13 (Line 245)

“Six studies reported clinical presentation with 61 of 175 infants (35%) having meningitis (versus 27% in the current study).”

Please provide the relevant references for this comparison.

Comment 14 (Table 2, Line 259)

“up to day 89”

Please revise to “day 7 to 89” for clarity.

Comment 15 (Table 2)

Does the “χ(2)” in the rightmost item of Zhang's (China, 2019 [15]) row refer to “χ²”? Please clarify the notation and make corrections as necessary.

Comment 16 (Table 2, Line 258)

“3 day 0–2 Number of cases of EOD and LOD may not be totally accurate as difficult to derive from data provided”

Please revise to:

“The number of EOD and LOD cases may not be accurate, as they are difficult to derive from the provided data.”

Comment 17 (Table 2, lines 260–261)

The description beginning with “5 Two had prior LOD …” refers to an item labeled “5”; however, I am unable to identify where “5” is shown in Table 2. Please clarify the correspondence between this description and the table entries.

Comment 18 (Table 2, line 262)

The text states that “6 appears to overlap with patients reported in the Georget-Bouquin study [21].” However, the Georget-Bouquin study [21] is not cited elsewhere in the manuscript, either in Table 2 or in the main text. Please clarify this reference and revise accordingly.

Comment 19 (Line 263)

Was the review of VLOD limited to articles published in English? Reference [21] appears to be a French-language publication. Please describe the methodology of the literature review, including the publication period searched, databases used, and languages included.

Comment 20 (References)

The capitalization of words in reference titles is inconsistent. Please revise all references so that they comply with the journal’s formatting guidelines.

Comment 21 (Reference 8)

8. 8. Millar JR, Anglemyer A, Werno A, Austin NC, Walls T. Epidemiology of

340 Infant Group B Streptococcus Infection in New Zealand: A 10-Year Retrospective Study. Pediatr Infect Dis J. 2025.

Revise the duplication reference number. Additionally, include the number of volume and pages.

Comment 22 (Reference 24)

Takahara T, Matsubara K, Maihara T, Yamagami Y, Chang B. Ultra-late-onset Meningitis Caused by Serotype IX Group B Streptococcus. Pediatr Infect Dis J. 34. United States 2015. p. 801

Please revise the journal name for this reference.

Comment 23 (References)

References 25 and 26 could not be identified in the reference list. Please clarify or revise accordingly.

Reviewer #2: Thank you for revising your manuscript.

I have a few additional minor comments

Abstract Lines 57-60. I would edit this to align closer with your manuscript’s final statement :"GBS remains an important cause of infant morbidity and mortality in the first three months of life. VLOD occurring beyond this period is likely under‑recognized in routine surveillance, and our findings highlight the need for greater clinical awareness and further research into its epidemiology and underlying immunologic determinants."

Lines 87-89: "In the early years of the study, clinicians were just becoming aware of the benefits of GBS screening, so some did not screen and/or gave intrapartum antibiotics only in the presence of risk factors for GBS transmission" This new statement lacks clarity. It reads more like a personal opinion, rather than a data-driven statement. Please re-write.

Lines 229-231: I suggest you remove this. As I mentioned in my previous comments, EOD and LOD might share common risk factors, but the pathophysiology is different, high inoculum vs persistent colonisation.

**Do you want your identity to be public for this peer review?** For information about this choice, including consent withdrawal, please see our Privacy Policy

Reviewer #1: No

Reviewer #2: No

---

## [Author Response · Author response to Decision Letter 2]

14 Jan 2026

Reviewer #1: The authors conducted a retrospective analysis of the clinical and microbiological characteristics of late-onset disease (LOD) and very late-onset disease (VLOD) caused by group B streptococcus (GBS) over a 27-year period at four hospitals in Edmonton. While the revised manuscript has improved, the following points require clarification or further revision.

Major comments

Comment 1

The authors stated “The GBS prenatal protocol during most of these years included rectovaginal screening at 35 to 37 weeks gestation with intrapartum antibiotics for all who screened positive and for unscreened persons with risk factors for GBS transmission. In the early years of the study, clinicians were just becoming aware of the benefits of GBS screening, so some did not screen or gave intrapartum antibiotics only in the presence of risk factors for GBS transmission.”

As I indicated in my comments on the original manuscript, the authors should more clearly explain when the Canadian guidelines regarding rectovaginal culture screening and intrapartum antibiotic prophylaxis for the prevention of early-onset GBS disease were implemented during the study period. Relevant guideline documents should be cited to clarify changes in clinical practice over time. The authors’ explanation of the revised manuscript is ambiguous for readers.

Reply: Thanks for pointing out that we needed to provide the reader with more detail. We now state:” In 1997, the initial Society of Obstetricians and Gynecologists of Canada GBS guideline recommended either rectovaginal screening at 35 to 37 weeks gestation with intrapartum antibiotics for all who screened positive or no screening with intrapartum antibiotics for those with risk factors [7]. In 2004 their guideline was revised to recommend only the former approach [8].”

Comment 2

As I indicated in my comments on the original manuscript, the definition of CSF pleocytosis differs for infants <1 month and ≥1 month of age.

The authors’ response stating that “cerebrospinal fluid (CSF) pleocytosis was defined as >20 × 10⁹/L white blood cells based on the assumption that many infants would be preterm” is not scientifically justified. In cases of late-onset disease (LOD), the proportion of preterm births is less than half, and therefore this assumption does not adequately support the chosen threshold.

The authors should define the criterion used for CSF pleocytosis and revise the number of cases classified as possible meningitis accordingly. As a consequence, the conclusions of the manuscript may also require revision.

The authors are encouraged to review relevant studies, including but not limited to the following, and to cite literature that best supports the CSF reference values adopted in the present study:

- Zimmermann P, et al. Normal Values for Cerebrospinal Fluid in Neonates: A Systematic Review. Neonatology. 2021.

- Kestenbaum LA, et al. Defining cerebrospinal fluid white blood cell count reference values in neonates and young infants. Pediatrics. 2010.

Reply: We agree that we need to justify the definition that we use. We reviewed the literature and chose to use the Kastenbaum study results as it was a large study and one of the few to report results beyond the neonatal period. The 95th percentile in that study was 19 WBCs in the first 28 days of life and 9 WBCs on days 29 to 56. We took the liberty of rounding each up to a round number and now state: “CSF pleocytosis was defined as > 20 X 106/L WBCs up to 28 days of age > 10 X 106/L WBCs at days 29 to 56 [8] and > 5 X 106/L WBCs after 56 days of age.”

This does not change our results as the three infants with 6 to 20 × 106/L WBC’s after 30 days of age had 6, 9 and 9 WBCS respectively at 29 to 56 days of age.

Comment 3

The unit used for the CSF WBC count appears to be incorrect. CSF WBC counts are expressed as ×10⁶/L (equivalent to cells/µL). Accordingly, the statement defining CSF pleocytosis as “>20 ×10⁹/L WBCs” is a critical error. This is not merely a semantic issue, as the definition of pleocytosis directly affects case classification.

Reply: Thanks for pointing out this error which has now been corrected throughout the manuscript.

Comment 4

In response to my comment that the original manuscript should clarify what historic VLOD cases reported in the literature refers to, the authors state that they reviewed VLOD cases but not LOD cases in the literature. They note that reviewing previously reported LOD cases would be a daunting task. While this explanation is understandable from a practical standpoint, it does not fully align with the stated objective of the study.

The authors describe their objectives as “to describe and compare the incidence and characteristics of LOD and VLOD in a cohort of neonates in a Canadian city over a prolonged period”. Based on this objective, it is unclear why the literature review focuses exclusively on VLOD, and why previously reported data are not addressed in the same way for LOD.

If the authors believe that a focused review of VLOD cases is particularly warranted, this rationale should be stated in the manuscript. Specifically, an explicit explanation should be added before the section reviewing VLOD cases in the Discussion to clarify why only VLOD is reviewed and how this approach supports the overall aims of the study.

Reply: Thanks for suggesting how to fix this awkward transition in the discussion. We now state: “Even less is known about the source of GBS or the characteristics of VLOD. Because of this, we completed a narrative review of the literature for VLOD case series. This identified 352 cases in 13 studies dating back to 1995 (Table 2).”

Comment 5

Related to Comment 4, the presentation of serotype distribution in Figure 2 appears incomplete. Showing only previously reported VLOD cases, without corresponding data for LOD, provides an unbalanced view and does not fully align with the overall study objective.

I therefore suggest removing the previously reported VLOD data from Figure 2. Instead, a brief comparison of serotype distributions between the present study and previously reported VLOD cases could be more appropriately addressed in the Discussion.

Reply: We have removed the previous VLOD cases from Figure 2 and now provide that data in the Discussion.

Comment 6 (Statistical analysis)

In the Abstract, main text, and tables, the authors report comparisons such as “n = 3 (27%) versus n = 72 (65%); p = 0.036” and “n = 5 (45%) versus n = 17 (15%); p = 0.039.” Based on these percentages, it appears that the denominators may be 11 and 111, respectively; however, this is not explicitly stated and should be clarified.

Reply: Thanks for the suggestion. Denominators were added where appropriate in the Abstract and Results.

In the Methods section, the authors indicate that Fisher’s exact test was used when the expected cell frequency was <5 in any cell of a 2×2 table. Therefore, it is assumed that Fisher’s exact test was applied to these comparisons. However, when I applied Fisher’s exact test using my standard statistical software, however, the reported p values could not be reproduced.

To ensure transparency and reproducibility, the authors are requested to provide the underlying 2×2 contingency tables and the corresponding results of Fisher’s exact test for these comparisons. It would be sufficient to provide this information in the response to the reviewers. Concomitantly, other statistical analyses are required to be re-examined.

Reply: See below for the 2x2 (or RxC) comparisons with associated p-values in Table 1. As stated in methods, we used www.openepi.com (an open source software) for comparisons. Fisher’s exact test was used when the expected cell frequency was <5 in any cell of a 2×2 table (highlighted in yellow for the outputs pasted in the document).

Supplementary Material – 2x2 or RxC tables with Chi Square analyses for data presented in Table 1

Results from OpenEpi, Version 3, open source calculator—Available at www.openepi.com

(Data presented in the order the comparisons appear in the table)

Maternal GBS Colonization

LOD VLOD

Y 39 2 41

Mat_GBS_Col N 42 4 46

Unk 30 5 35

111 11 122

Chi Square for R by C Table

Chi Square= 2.046

Degrees of Freedom= 2

p-value= 0.3595

2 x 2 Table Statistics

Previous Infant with GBS infection

LOD VLOD

Y 2 0 2

Prev_Infant_GBS No 108 11 119

110 11 121

Chi Square and Exact Measures of Association

Test Value p-value(1-tail) p-value(2-tail)

Uncorrected chi square 0.2034 0.3260 0.6520

Yates corrected chi square 0.6228 0.2150 0.4300

Mantel-Haenszel chi square 0.2017 0.3267 0.6534

Fisher exact 0.8258 >0.9999999

Mid-P exact 0.4129 0.8258

Maternal Diabetes/GDM

Disease

LOD VLOD

Y 9 1 10

Mat_GDM No 101 10 111

110 11 121

Chi Square and Exact Measures of Association

Test Value p-value(1-tail) p-value(2-tail)

Uncorrected chi square 0.0109 0.4584 0.9168

Yates corrected chi square 0.2207 0.3192 0.6385

Mantel-Haenszel chi square 0.01081 0.4586 0.9172

Fisher exact 0.6293(P) >0.9999999

Mid-P exact 0.4275(P) 0.8550

Corticosteroids prior to birth

LOD VLOD

Y 30 3 33

Antenatal_CS No 80 8 88

110 11 121

Chi Square and Exact Measures of Association

Test Value p-value(1-tail) p-value(2-tail)

Uncorrected chi square 0 0.5000 >0.9999999

Yates corrected chi square 0.126 0.3613 0.7226

Mantel-Haenszel chi square 0 0.5000 >0.9999999

Fisher exact 0.6220(P) >0.9999999

Mid-P exact 0.4846(P) 0.9693

Vaginal Delivery

LOD VLOD

(+) 72 3 75

Vag_Del (-) 39 8 47

111 11 122

Chi Square and Exact Measures of Association

Test Value p-value(1-tail) p-value(2-tail)

Uncorrected chi square 5.972 0.007268 0.01454

Yates corrected chi square 4.49 0.01705 0.03409

Mantel-Haenszel chi square 5.923 0.007473 0.01495

Fisher exact 0.01812 0.03623

Mid-P exact 0.01055 0.02110

Male sex

Disease

LOD VLOD

Y 47 3 50

Male No 63 8 71

110 11 121

Chi Square and Exact Measures of Association

Test Value p-value(1-tail) p-value(2-tail)

Uncorrected chi square 0.985 0.1605 0.3210

Yates corrected chi square 0.4508 0.2510 0.5020

Mantel-Haenszel chi square 0.9769 0.1615 0.3230

Fisher exact 0.2549 0.5097

Mid-P exact 0.1732 0.3464

Preterm

Disease

LOD VLOD

Y 50 8 58

Preterm N 58 3 61

108 11 119

Chi Square and Exact Measures of Association

Test Value p-value(1-tail) p-value(2-tail)

Uncorrected chi square 2.791 0.04739 0.09479

Yates corrected chi square 1.834 0.08794 0.1759

Mantel-Haenszel chi square 2.768 0.04809 0.09619

Fisher exact 0.08715(P) 0.1743

Mid-P exact 0.05444(P) 0.1089

Mastitis prior to GBS onset

Disease

LOD VLOD

Y 1 0 1

Mastitis_prior No 109 11 120

110 11 121

Chi Square and Exact Measures of Association

Test Value p-value(1-tail) p-value(2-tail)

Uncorrected chi square 0.1008 0.3754 0.7508

Yates corrected chi square 2.042 0.07657 0.1531

Mantel-Haenszel chi square 0.1 0.3759 0.7518

Fisher exact 0.9091 >0.9999999

Mid-P exact 0.4545 0.9091

Mechanical ventilation during birth hospitalization

Disease

LOD VLOD

Y 17 5 22

Mech_Vent No 94 6 100

111 11 122

Chi Square and Exact Measures of Association

Test Value p-value(1-tail) p-value(2-tail)

Uncorrected chi square 6.151 0.006568 0.01314

Yates corrected chi square 4.281 0.01928 0.03855

Mantel-Haenszel chi square 6.1 0.006758 0.01352

Fisher exact 0.02689(P) 0.05378

Mid-P exact 0.01570(P) 0.03141

Please note: Yates corrected chi square value is quoted in the table as no cell number is <5

Necrotising Enterocolitis (NEC)

Disease

LOD VLOD

Y 5 1 6

NEC No 105 10 115

110 11 121

Chi Square and Exact Measures of Association

Test Value p-value(1-tail) p-value(2-tail)

Uncorrected chi square 0.4384 0.2539 0.5079

Yates corrected chi square 0.004384 0.4736 0.9472

Mantel-Haenszel chi square 0.4348 0.2548 0.5097

Fisher exact 0.4427(P) 0.8854

Mid-P exact 0.2676(P) 0.5351

Serious bacterial infection with bacteria other than GBS during GBS admission

LOD VLOD

Y 4 3 7

Inf_other Bac No 106 8 114

110 11 121

Chi Square and Exact Measures of Association

Test Value p-value(1-tail) p-value(2-tail)

Uncorrected chi square 10.25 0.0006834 0.001367

Yates corrected chi square 6.372 0.005796 0.01159

Mantel-Haenszel chi square 10.17 0.0007155 0.001431

Fisher exact 0.01626(P) 0.03252

Mid-P exact 0.008716(P) 0.01743

Hospital status at GBS onset

LOD VLOD

Adm_from home 77 9 86

Adm_during Birth Hospitalization 34 2 36

111 11 122

Chi Square and Exact Measures of Association

Test Value p-value(1-tail) p-value(2-tail)

Uncorrected chi square 0.7456 0.1939 0.3879

Yates corrected chi square 0.2673 0.3026 0.6052

Mantel-Haenszel chi square 0.7395 0.1949 0.3898

Fisher exact 0.3147(P) 0.6295

Mid-P exact 0.2115(P) 0.4230

Comment 7 (Introduction)

In the Introduction, only reference #1 is cited. This is somewhat unexpected, as the Introduction is intended to summarize both what is already known and what remains unknown in the field. To provide appropriate context for the study, it would be helpful to include additional key references that reflect the existing body of literature.

Incorporating multiple relevant citations would strengthen the background section and better support the rationale for the present study.

Reply: More information was added on LOD and VLOD presentation and pathogenesis including more references.

Comment 8 (Table 1)

The footnotes to Table 1, under “Invasive infection with bacteria other than GBS during GBS admission,” list the entries

“5 Urinary tract infection (N = 1)” and “6 Urinary tract infection (N = 1)”.

However, a urinary tract infection is generally not typically classified as an invasive bacterial infection. Therefore, these entries should therefore be removed from this category, and the corresponding statistical analyses should be revised. Any affected results should also be corrected in the Abstract and the main text.

Reply: Thanks for pointing out this mistake. We agree with the reviewer that UTIs in infants a

---

## [Decision Letter · Decision Letter 2]

10 Feb 2026

Shifting From an Expected to an Opportunistic Pathogen: Characteristics and Trends of Infant Late and Very Late Onset Group B streptococcal (GBS) infection in a Canadian city over a 27-year period

PONE-D-25-58766R2

Dear Dr. Robinson,

We’re pleased to inform you that your manuscript has been judged scientifically suitable for publication and will be formally accepted for publication once it meets all outstanding technical requirements.

Kind regards,

Yung-Fu Chang

Academic Editor

PLOS One

Additional Editor Comments (optional):

Reviewers' comments:

Reviewer's Responses to Questions

**Comments to the Author**

Reviewer #1: All comments have been addressed

Reviewer #2: All comments have been addressed

2. Is the manuscript technically sound, and do the data support the conclusions?

Reviewer #1: Yes

Reviewer #2: Yes

3. Has the statistical analysis been performed appropriately and rigorously?

Reviewer #1: Yes

Reviewer #2: Yes

4. Have the authors made all data underlying the findings in their manuscript fully available?

Reviewer #1: Yes

Reviewer #2: Yes

5. Is the manuscript presented in an intelligible fashion and written in standard English?

Reviewer #1: Yes

Reviewer #2: Yes

Reviewer #1: I thank the authors for their responses to the comments. The manuscript has been well revised, and the authors have satisfactorily addressed the issues raised during the review process.

Reviewer #2: I have no further substantive comments for the manuscript. Overall, I am satisfied that the manuscript meets the required standards and I am happy to recommend it for publication.

**Do you want your identity to be public for this peer review?** For information about this choice, including consent withdrawal, please see our Privacy Policy

Reviewer #1: No

Reviewer #2: No

---

## [Editor Report · Acceptance letter]

PONE-D-25-58766R2

PLOS One

Dear Dr. Robinson,

I'm pleased to inform you that your manuscript has been deemed suitable for publication in PLOS One. Congratulations! Your manuscript is now being handed over to our production team.

Kind regards,

on behalf of

Dr. Yung-Fu Chang

Academic Editor

PLOS One